

# Vivid-Panda
## Internetowy System Informatyczny Umożliwiający Edycję Grafiki

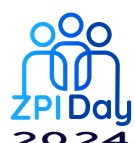

**Autors**: Katsiaryna Viarenich [1] · Karolina Majkowska [2] · Uladzimir Mazaleuski [3] · Paweł Dudek[4]

**Supervisor:** Martin Tabakow

### Abstract

Celem naszego zespołowego przedsięwzięcia inżynierskiego było stworzenie internetowego systemu umożliwiającego edycję grafiki, który pozwalałby na zarówno proste operacje, jak i te zaawansowane, które są wsparte modelami sztucznej inteligencji. Stworzona przez nas aplikacja oferuje narzędzia, które pozwalają na tradycyjne zmiany jak przycięcie zdjęcia, jego obrót czy nałożenie różnych filtrów, ale też o wiele ciekawsze efekty, jak koloryzacja grafiki, poprawa jej jakości czy usunięcie albo dodanie elementu do zdjęcia. Może mieć to duże znaczenie dla osób, które interesują się edycją grafiki i zajmują się nią profesjonalnie czy też takich, które dopiero zaczynają swoją przygodę z nią, albo potrzebują jednorazowo skorzystać z takiego rozwiązania. Dzięki niskiemu progowi wejścia do aplikacji, jest ona dostępna i przyjazna dla każdego, co jest ważne, zwłaszcza dla początkującego użytkownika.

## 1 WPROWADZENIE

Edycja zdjęć za pomocą sztucznej inteligencji stała się bardzo popularna ze względu na jej bardzo szybki rozwój. Istniejące na rynku modele oferują coraz większą liczbę funkcji i zapewniają coraz realistyczniejsze wyniki. Jednakże na rynku istnieją jedynie rozwiązania, które nie łączą ze sobą wielu funkcji na raz, są trudne w użyciu i mają wysoki próg wejścia lub są niedostępne dla wielu ze względu na wysokie wymagania sprzętowe czy nieprzystępną cenę.

Ta nisza była motywacją na stworzenie aplikacji **Vivid-Panda**, łączącej w sobie wiele popularnych i potrzebnych użytkownikom funkcji opartych na rozwiązaniach AI takich jak:

· **koloryzacja czarno-białych zdjęć**, pozwalająca na przywrócenie dawnego blasku wspomnieniom z przeszłości;

· **usuwanie i dodawanie elementów na zdjęciach**, umożliwiająca na kontrolowanie środowiska na zdjęciu już po jego wykonaniu;

· **zwiększanie rozdzielczości**, poprawiającą szczegółowość zdjęcia, co ujawni wcześniej niezauważalne szczegóły;

Wszystkie te funkcje dostępne są za pomocą nowoczesnego, zrozumiałego interfejsu.

Dzięki takiemu podejściu grupa użytkowników aplikacji nie została ograniczona jedynie do profesjonalistów, chcących poświęcić czas na naukę narzędzia. Rozwiązanie jest zachęcające również dla laików technologicznych czy osób chcących szybko i prosto uzyskać pożądane wyniki.

Ponadto rozwarstwienie aplikacji dzięki architekturze mikroserwisów pozwoli na bezproblemowe wdrażanie coraz nowszych i skuteczniejszych modeli sztucznej inteligencji bez ingerencji w pracę użytkownika.

Dalekosiężnymi celami biznesowymi projektu jest więc trafienie do jak największej liczby użytkowników, pozwalając na rozwijanie modeli sztucznej inteligencji. Poprzez korzystanie z danych, które generują użytkownicy, będzie możliwe doszkalanie/tworzenie modeli. Osiągając ten cel stworzone zostanie również stałe źródło dochodów, poprzez wyświetlanie reklam czy potencjalny model freemium.

[1] ORCID: 0000-0001-9789-3404
[2] ORCID: 0009-0002-1249-3001
[3] ORCID: 0009-0000-5154-7389
[4] ORCID: 0009-0004-7688-1403

## 2 PRZEGLĄD ISTNIEJĄCYCH ROZWIĄZAŃ

Na rynku możemy znaleźć już wiele aplikacji, które umożliwiają użytkownikowi na edycję grafiki. Niektóre z nich oferują nawet zaawansowane rozwiązania wsparte sztuczną inteligencją. Jednak żadne ze znalezionych rozwiązań nie było w pełni zadowalające z perspektywy wszystkich wymagań użytkowników: idealna aplikacja do edycji zdjęć powinna mieć prosty interfejs, być bezpłatna, charakteryzować się niskimi wymaganiami systemowymi, a jednocześnie oferować zaawansowane funkcje oparte na AI. Porównanie istniejących narzędzi aktualnie dostępnych na rynku pokazuje tabela 1.

| | Photoshop | Canva | Photopea | Luminar Neo | Pixlr | Fotor | GIMP | Snapseed | Topaz AI | Affinity Photo | **Vivid-Panda(ours)** |
|---|---|---|---|---|---|---|---|---|---|---|---|
| Prosty interfejs | ✗ | ✓ | ✗ | ✗ | ✓ | ✓ | ✗ | ✓ | ✗ | ✗ | ✓ |
| Bezpłatny/Freemium | ✗ | ✓ | ✓ | ✗ | ✓ | ✓ | ✓ | ✓ | ✗ | ✗ | ✓ |
| Zaawansowane funkcjonalności AI | ✓ | ✗ | ✗ | ✓ | ✗ | ✗ | ✗ | ✗ | ✓ | ✓ | ✓ |
| Niskie wymagania systemowe | ✗ | ✓ | ✓ | ✗ | ✓ | ✓ | ✗ | ✗ | ✗ | ✗ | ✓ |

Table 1: Porównanie aplikacji do edycji zdjęć

Te proste porównanie pokazuje, że nasze rozwiązanie spełnia wszystkie zauważone potrzeby użytkowników. Czyni to naszą aplikację konkurencyjną na rynku podobnych systemów.

## 3 TECHNOLOGIE

Zastosowanie technologii webowych, pozwala na zmniejszenie wymagań sprzętowych czego skutkiem jest zwiększenie dostępności aplikacji. W projekcie wykorzystane zostały modele AI open-source, co umożliwia zaawansowaną edycję bez konieczności tworzenia nowych zbiorów danych. Dodatkowo, obniżony próg wejścia zapewniony jest dzięki intuicyjnemu i ergonomicznemu interfejsowi.

Sam wybór języka Python i frameworka Flask oparty był na szerokim wsparciu dla potrzebnych funkcji i dobrze rozwiniętej społeczności. Dodatkowo w związku z początkowym założeniem architektury monolitu (które uległo zmianie po zauważeniu problemu skalowalności modułu AI) postanowiliśmy zostać przy tym rozwiązaniu ze względu na jego lekkość i w.w. atuty. Jako bazę danych wybraliśmy MongoDB, które w prosty sposób pozwala na przechowywanie plików. Sposób ich przechowywania przyspiesza oraz udostępnia ustandaryzowany interfejs umożliwiający dostęp do nich.

Wybór Next.js i języka Typescript jako technologii frontendowych został podjęty ze względu na mnogości gotowych bibliotek oraz wsparciu dla dynamiczności strony. Użyliśmy również MaterialUI i Tailwind dla stworzenia czytelnego i zrozumiałego wyglądy strony.

Do projektu użyte też zostały inne, pomocnicze technologie. Wspierały one proces tworzenia oprogramowania, jego dokumentację oraz wdrożenie. Dzięki nim cały system jest kompletny oraz odpowiednio udokumentowany.

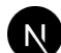 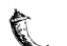 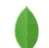 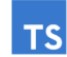 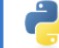 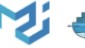 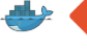 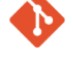 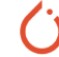 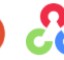 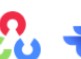 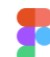 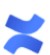 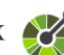 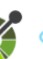 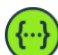

## 4 WYNIKI

### 4.1 Struktura i działanie aplikacji

Udało się zaimplementować wszystkie założone funkcje naszego systemu. Wdrożone przypadki użycia znajdują się na diagramie nr 1

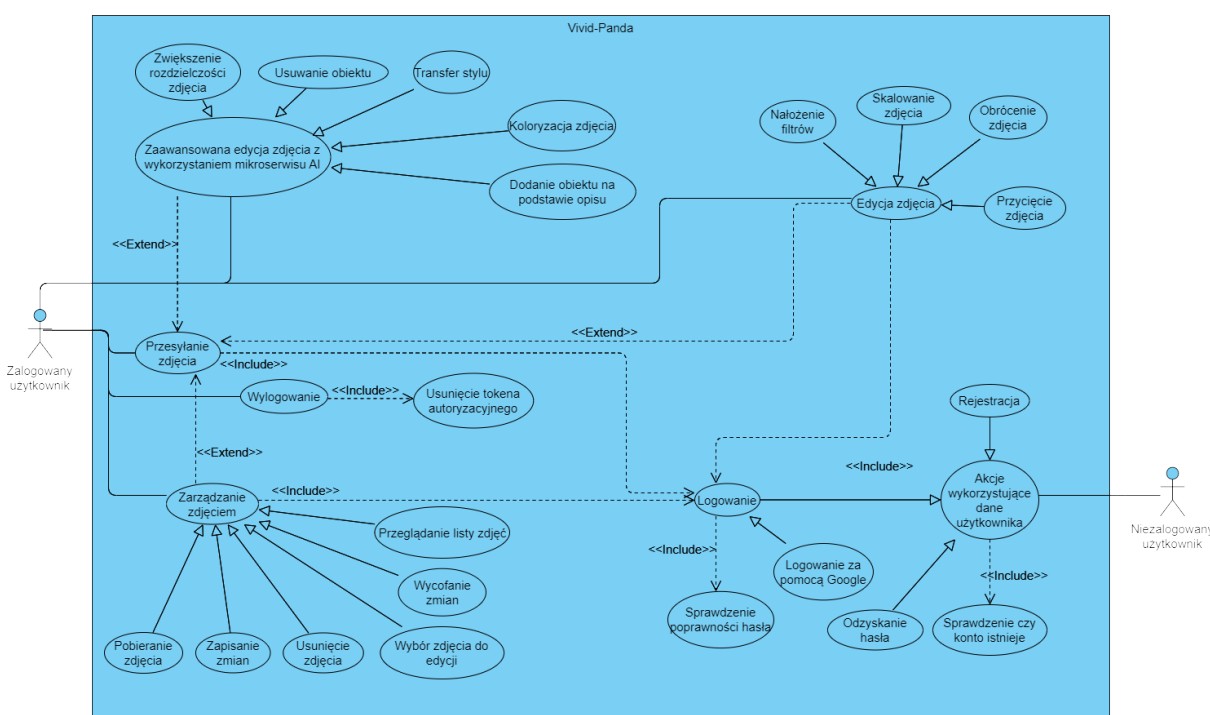

Figure 1: Diagram przypadków użycia

Jednym z głównych założeń aplikacji było stworzenie czytelnego i intuicyjnego interfejsu użytkownika. Część interfejsu użytkownika została zaprezentowana poniżej na obrazkach nr 2, 3, 4.

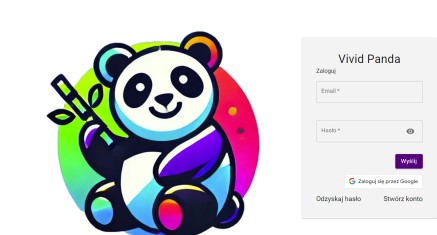

Figure 2: Ekran logowania

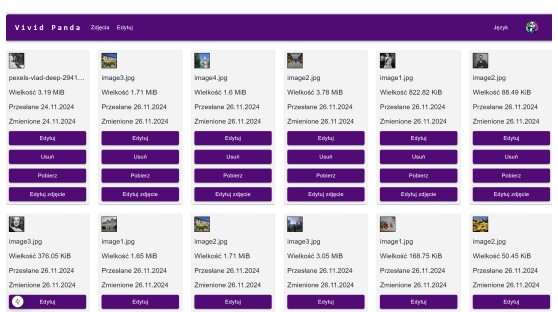

Figure 3: Ekran listowania zapisanych zdjęć

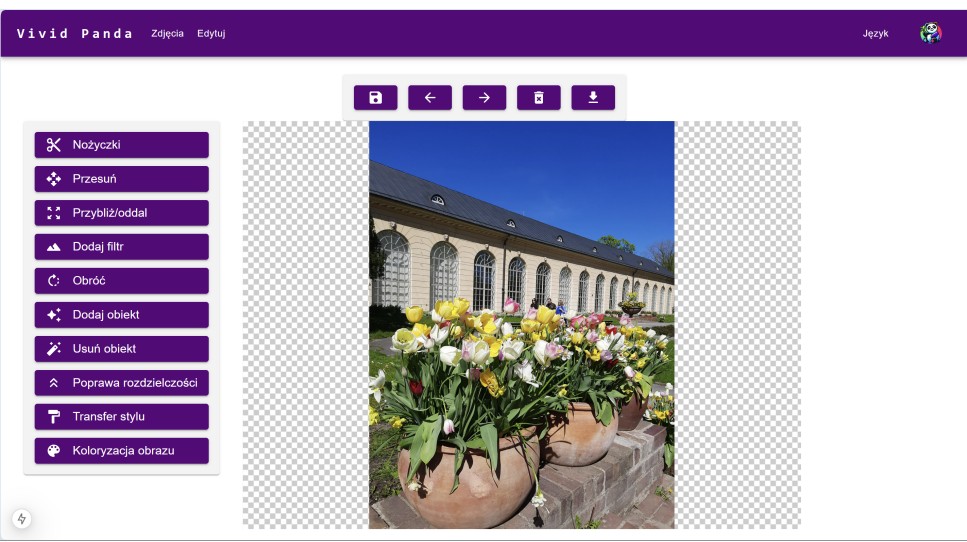

Figure 4: Ekran edytora aplikacji z własnoręcznie wykonanym zdjęciem

Architektura naszego systemu składa się z części: frontend, backend, baza danych oraz mikroserwis z rozwiązaniami sztucznej inteligencji. Jest to klasyczne podejście do tworzenia aplikacji webowych (frontend, backend), a AI jako mikroserwis zapewnia nam elastyczność i skalowalność użytych modeli. Szczegóły zostały zaprezentowane na diagramie nr 5.

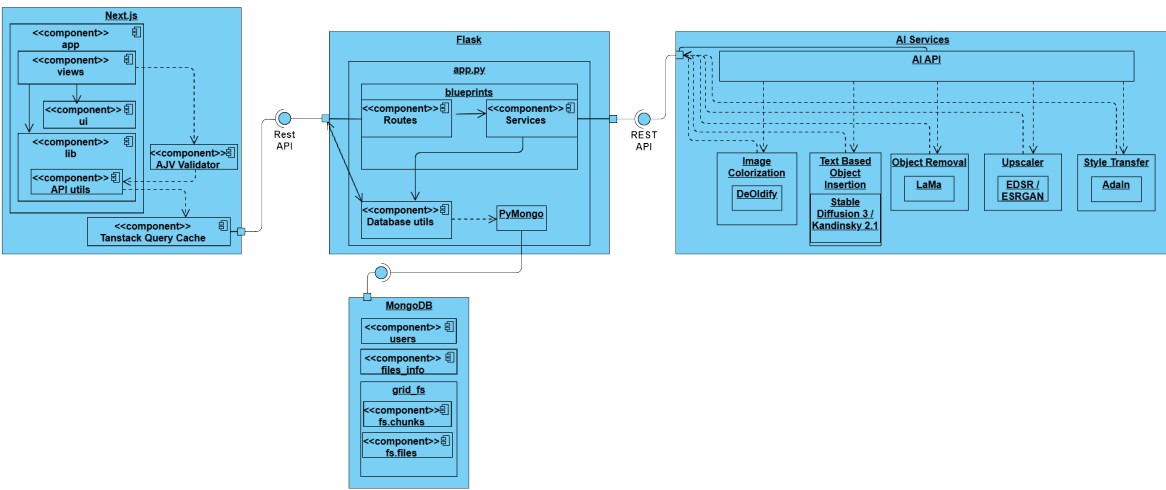

Figure 5: Diagram komponentów wytworzonego systemu informatycznego

Biorąc zatem pod uwagę fakt zrealizowania wszystkich założonych funkcjonalności, powstał kompletny system do edycji grafiki z możliwością rozbudowy go o coraz to nowsze narzędzia. Możemy zatem stwierdzić, że osiągnięty został nasz główny cel biznesowy, czyli dostarczenie użytkownikowi prostej w obsłudze oraz w pełni funkcjonalnej aplikacji do obróbki grafiki. Nasz system zapewnia narzędzia, które spełniają różne potrzeby użytkowników. Wraz z jej rozwojem dodawane mogą być oraz to nowsze opcje, dzięki którym aplikacja może stać się konkurencją dla wielu profesjonalnych, dostępnych na rynku programów.

Pod kątem technicznym udało się również osiągnąć założone przez nas cele. Stworzona aplikacja zapewnia użytkownikowi prostotę obsługi oraz jak najnowocześniejsze technologie. Użyte modele sztucznej inteligencji również zostały wybrane tak, aby zwracać jak najlepsze rezultaty. Aplikacja jest również łatwo rozszerzalna o nowe funkcje czy nowsze modele. Dzięki użyciu znanych frameworków i wzorców projektowych, udało nam się uzyskać aplikację, która jest nowoczesna technologicznie i łatwa w ustrzymaniu.

Dodatkowo, jako że bardzo ważnym aspektem aplikacji webowej jest jej wygląd, warto wspomnieć, że kolorystyka została dobrana w taki sposób, aby pomimo przyjemnego dla oka wyglądu charakteryzowała się również wysoką dostępnością. W tym celu użyta została specyfikacja WCAG (*Web Content Accessibility Guidelines*). Wszystkie nachodzące na siebie kolory zostały zweryfikowane [16] i osiągają wartość kontrastu co najmniej *7:1* co oznacza, że spełnione są wymagania postawione przez specyfikację.

Użyte zostało też oprogramowanie *Vega Vulnerability Scanner* [12], które nie wykazało żadnych błędów programistycznych. Wszystkie znalezione potencjalne luki okazały się być spowodowane dynamicznym charakterem stron. W aktualnej wersji aplikacji błędy te nie są krytyczne i nie powodują problemów bezpieczeństwa dla użytkownika i twórców. Wraz z rozwojem naszego rozwiązania, niedoskonałości wykryte przez oprogramowanie VEGA będą naprawione.

Również technologie zostały dobrane tak, aby legalnym było ich użycie w celach komercyjnych bez konieczności dodatkowych kosztów pozyskania licencji. Wszystkie użyte biblioteki, frameworki oraz języki udostępnione są na opensourcowych licencjach, głównie MIT oraz Apache. Pozwala to nam na bezpieczne tworzenie oprogramowania, które możemy wprowadzić na rynek, oraz używać bez naruszania praw autorskich.

W celu zapewnić bezpieczeństwo danych użytkownika oraz niezawodność naszej aplikacji przeprowadzone zostały różnego rodzaju testy. Testy jednostkowe nie pokrywają kodu w pełni, ponieważ moduł AI nie może być nimi pokryty, ze względu na brak określonych wymagań i możliwą różnicę w wynikach. Jednakże testy akceptacyjne, integracyjne oraz obciążeniowe dały nam zadowalające wyniki. Wykazało to, że aplikacja spełnia nasze założenia techniczne oraz jest bezpieczna dla użytkownika i jego danych.

| Testy jednostkowe | Testy integracyjne | Testy obciążeniowe | Testy akceptacyjne |
|---|---|---|---|
| Pokrycie 61% | Pokrycie 100% | Spełnia wymagania | 15/15 |

Table 2: Wykonane testy

Udało się osiągnąć te cele pomimo ograniczeń czasowych (projekt musiał zostać zrealizowany w przeciągu dwóch miesięcy) i braku wysokiej klasy kart graficznych (które są niezbędne do inferencji dużych modeli sztucznej inteligencji). Kluczem do naszego sukcesu była ciężka praca oraz znalezienie wydajnych rozwiązań AI, które cechują się niskimi wymaganiami sprzętowymi, a jednocześnie zachowują wysoką jakość.

## 4.2  Moduł AI

Podczas implementacji modułu AI naszym głównym celem było stworzenie rozwiązania, które zapewni wysoką jakość wyników przy minimalnym wykorzystaniu zasobów. Dążono do opracowania systemu działającego zarówno na laptopach, jak i skalowalnego do pracy na serwerach, z wykorzystaniem nowoczesnych technologii i modeli sztucznej inteligencji.

*Do koloryzacji zdjęć* zastosowano model DeOldify [6], który opiera się na zaawansowanym modelu GAN [4] z unikalnym podejściem NoGAN. Ta technologia umożliwia stabilne i efektywne uczenie, co przekłada się na wysoce realistyczne odwzorowanie kolorów i detali. Wykorzystanie architektury U-Net [10] z spectral normalization i self-attention [14], zapewnia precyzję i szczegółowość koloryzacji, a strata bazująca na feature loss (VGG16 [11]) w połączeniu z wkładem krytyka podnosi jakość generowanych obrazów. Dzięki temu DeOldify pozwala na wydajną koloryzację zdjęć, która spełnia profesjonalne standardy jakości, jednocześnie zachowując umiarkowane wymagania sprzętowe.

*Do usuwania obiektów* wykorzystano model LaMa [13], który obecnie uznawany jest za jedno z najbardziej zaawansowanych rozwiązań w dziedzinie inpaintingu. Model wykorzystuje szybkie konwolucje Fouriera (Fast Fourier Convolutions, FFC [1]), dzięki czemu efektywnie zachowuje globalny kontekst obrazu. Jego szerokie pole recepcyjne i trening na dużych maskach umożliwiają skuteczne uzupełnianie nawet dużych braków w obrazie. Wyniki LaMa wyróżniają się wysoką spójnością i jakością, dzięki czemu model ten sprawdza się zarówno w prostych, jak i bardziej złożonych zadaniach.

*Do generowania obiektów* w obrazach wybraliśmy modele Stable Diffusion 3 [2] i Kandinsky 2.1 [9]. Stable Diffusion 3 to duży model generatywny, który zapewnia wysoką jakość dodawanych obiektów, szczególnie w kontekście bardziej złożonych scen. Dzięki swojej wszechstronności i dużym możliwością generacyjnym, jest idealnym rozwiązaniem, gdy wymagamy precyzyjnego dodawania nowych elementów w obrazie. Z kolei Kandinsky 2.1 został wybrany z powodu swojej lekkości i wydajności w środowiskach lokalnych. Jego cechą charakterystyczną jest generowanie stylizowanych, artystycznych efektów, co czyni go odpowiednim do obrazów wymagających kreatywnych i mniej realistycznych rezultatów.

*Stylizację obrazów* zrealizowano za pomocą technologii AdaIN (Adaptive Instance Normalization [5]), która umożliwia dynamiczne dopasowanie stylu do zawartości sceny. Rozwiązanie to cechuje się wysoką jakością generowanych zdjęć i możliwością przetwarzania w czasie rzeczywistym, co czyni je bardziej wydajnym w porównaniu do innych metod, takich jak Strotss [7], CycleGAN [17] czy klasyczny Neural Style Transfer [3], które również zostały przetestowane przez nas zespół.

*W celu zwiększania rozdzielczości* zdjęć użyto dwóch różnych modeli. EDSR (Enhanced Deep Super-Resolution [8]) to model, który zapewnia optymalny balans między jakością wyników, a obciążeniem sprzętowym, dzięki czemu sprawdza się w środowiskach lokalnych. ESRGAN (Enhanced Super-Resolution GAN [15]) z kolei charakteryzuje się nieco wyższą jakością na benchmarkach, ale wymaga większych zasobów obliczeniowych, co czyni go bardziej odpowiednim do zastosowań serwerowych.

Dzięki przemyślanemu doborowi technologii moduł AI spełnia wszystkie założenia projektu, oferując elastyczność, wysoką jakość i możliwość dostosowania w zależności od potrzeb użytkownika i dostępnej infrastruktury.

## 4.3 Architektura modułu AI

Architektura modułu AI, w której zaimplementowano opisane technologie, opiera się na kilku wzorcach projektowych. Centralnym elementem jest wzorzec Polecenie, który umożliwia wywołanie jednej z pięciu funkcjonalności modułu. W przypadku złożonych operacji, takich jak dodawanie obiektów, zastosowano również wzorzec Strategia, pozwalający na wybór odpowiedniego modelu (np. Stable Diffusion lub Kandinsky) w zależności od dostępnych zasobów. Dodatkowo, wzorzec Dekorator został wykorzystany do spójnego odczytu danych wejściowych, a Fasada zapewnia ujednolicenie procesu przetwarzania dla każdej z funkcjonalności. W celu uniknięcia wielokrotnego ładowania ciężkich modeli, był również zastosowany wzorzec projektowy Singleton.

Na rysunku nr.6 zaprezentowano przykładowe wyniki działania opisanych wcześniej modeli.

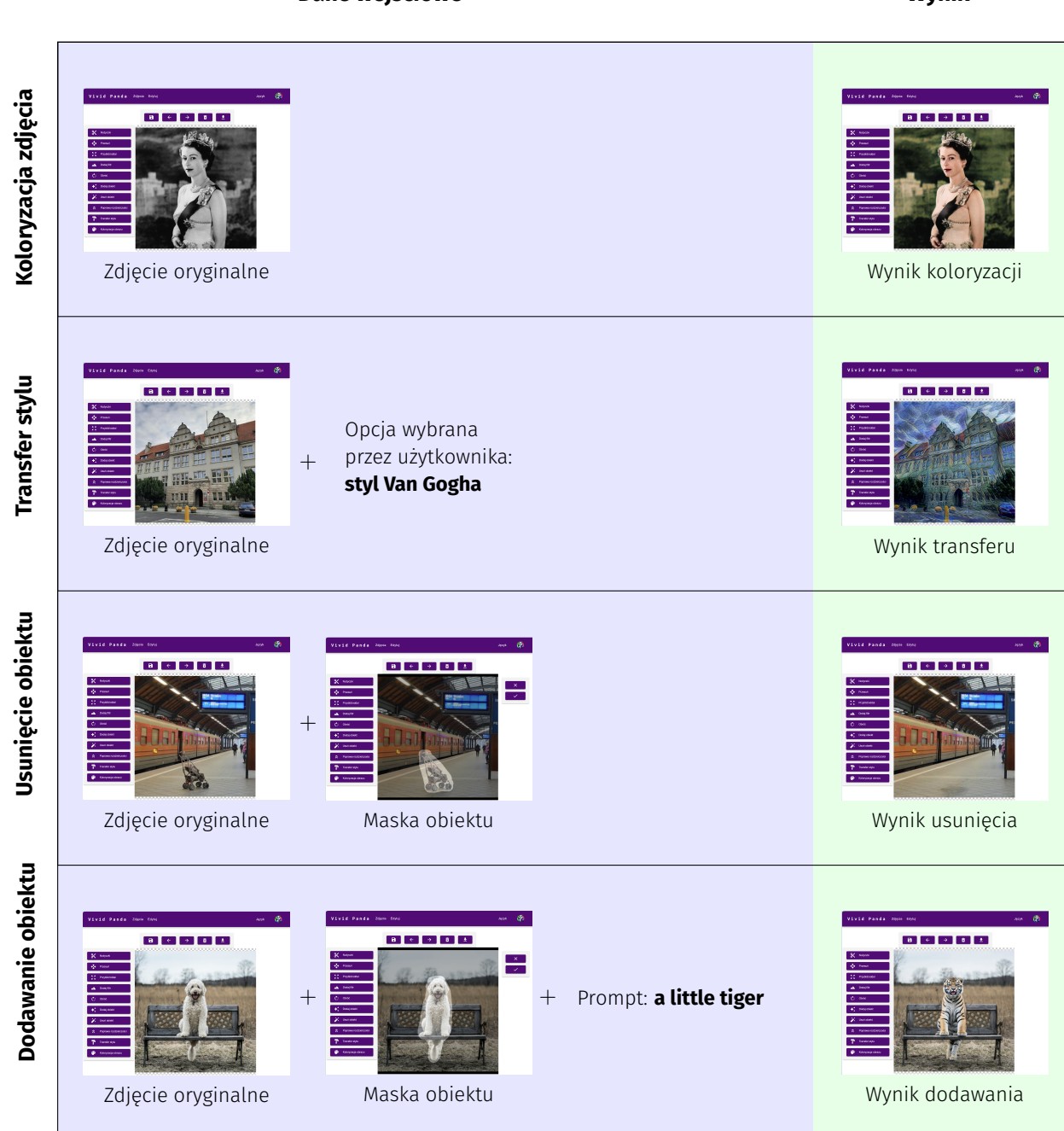

Figure 6: Przykładowe wyniki działania aplikacji z użyciem modułu AI.

# 5 WNIOSKI

## 5.1 Przyszłe kierunki rozwoju

Aplikacja może być rozwijana poprzez dodawanie coraz to nowszych funkcji edycji grafiki. Przewidujemy rozszerzenie o różne dodatkowe opcje wsparte sztuczną inteligencją, jak rozszerzanie obrazów czy automatyczna korekta zdjęcia, a także proste narzędzia, jak dodanie tekstu czy rysowanie. Oprócz samej edycji zdjęcia rozważane jest także dodanie takich opcji jak udostępnianie zdjęcia na portale społecznościowe czy możliwość pracy w grupach. Narzędzia te są coraz bardziej pożądane przez użytkowników aby efektywniej móc zarządzać swoimi zdjęciami i dzielić się nimi. Struktura naszego rozwiązania dodatkowo jest stworzona w taki sposób, że bez większego problemu jesteśmy w stanie wprowadzać coraz to bardziej dokładne modele sztucznej inteligencji oraz dodawanie nowych wraz z nowymi funkcjonalnościami. Jest to zaleta, która nie ogranicza nas przy rozwijaniu i skalowaniu naszej aplikacji.

Poza dodawaniem nowych opcji użytkownikowi, możliwy jest też rozwój w kierunku współprac z innymi programami od innych twórców. Zapewni to większą rozpoznawalność na rynku naszego systemu, co może przyciągnąć potencjalnych użytkowników. Współpraca te mogłaby się odbywać poprzez dostęp innych aplikacji do funkcji naszego rozwiązania.

Przyszły rozwój projektu ma sprawić, że stanie się on konkurencją dla profesjonalnych narzędzi do edycji grafiki. Powinien zostać jednak dostępny dla każdego, zatem jednym z planów na jej rozwój jest wprowadzenie modelu freemium, gdzie ze pewną opłatą jest dostęp do dodatkowych funkcji czy zwiększenie limitów aplikacji (jak liczba przesłanych zdjęć czy dostępność większej liczby opcji). Sama aplikacja zostałaby bezpłatna, jednak część jej funkcji dostępna byłaby dopiero po opłaceniu pewnej kwoty.

## 5.2 Podziękowania

Osoby, którym chcemy podziękować za wsparcie nas w realizacji projektu:
Nasz promotor: dr.inż. Martin Tabakow - specjalistyczna wiedza w dziedzinie sztucznej inteligencji, konstruktywne uwagi co do tworzenia aplikacji, korektę dokumentacji dr.inż. Piotr Syga - wiedza w kontekście tworzenia aplikacji webowych, bezpieczeństwa i dostępności

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
