# OpenReview forum: "Vivid-Panda"
_pwr.edu.pl/Wrocław_University_of_Science_and_Technology/2024/ZPI_Day — Wrocław University of Science and Technology 2024 ZPI Day Submission_

### Official Review · Reviewer_jwqj · 2024-12-03
**Interesting project but evaluation of results is marginal**

**Confidence:** 4
**Significance Of Results:** 3
**Overall Quality:** 3

**Compliance With Template:**

4: High Quality – The article contains all the required sections, which are well-written and substantively correct, although minor errors or shortcomings may be present. The overall structure is clear and coherent.

**Description Of Results:**

3: Average Quality – The results are described with moderate detail. Some examples or evaluation elements are present but insufficiently developed or incomplete.

**Feedback On Consistency:**

Most of the parts of the report are ok, except "Wyniki" section - it presents results of the work of the system only with a single figure (Fig. 6). The text around presents mostly the construction of the system (architecture and used technologies). The quality of the achieved results is not measured/presented.

The text is mostly consistent - but it looks strange that the main text is in Polish and "Table", "Figure", "Abstract", "References" are English words. Style and quality of the Polish language is appropriate - only minor errors can be found. Example problems:
- Autors => Authors / Autorzy
- "Te proste porównanie' => "To proste porównanie"
- "łatwa w ustrzymaniu" => "łatwa w utrzymaniu".
- "W celu zapewnić bezpieczeństwo danych użytkownika oraz niezawodność naszej aplikacji przeprowadzone zostały różnego rodzaju testy." => "W celu zapewnienia bezpieczeństwa danych użytkownika oraz niezawodności naszej aplikacji przeprowadzone zostały różnego rodzaju testy."
- "na-jbardziej" => "naj-bardziej" (niepoprawne miejsce przeniesienia)
- "dla stworzenia czytelnego i zrozumiałego wyglądy strony." => "dla stworzenia czytelnego i zrozumiałego wyglądu strony."

It can be problematic that Authors are using not precisely defined terms and claims which are not proven, e.g:
- "poprawiającą szczegółowość zdjęcia, co ujawni wcześniej niezauważalne szczegóły" (for sure in every case?)
- "przywrócenie dawnego blasku wspomnieniom z przeszłości" (how to measure "blask wspomnień"?)

**Potential For Development:**

Yes.

**Project Nature Evaluation:**

Yes, the project exhibits characteristics of an engineering work, with high level of utility, application of technical methods, and technological solutions.

**Technical Language Precision:**

4: High Quality – The language is appropriate for a technical report. Terminology is used correctly, and statements are precise, with only minor shortcomings that do not affect the overall clarity.

---

### Official Review · Reviewer_j3ZH · 2024-12-06
**Vivid-Panda**

**Confidence:** 2
**Significance Of Results:** 5
**Overall Quality:** 5

**Compliance With Template:**

5: Very High Quality – The article contains all the required sections, which are written in a very detailed, clear, and error-free manner. The structure is professional and meets expectations, and the content adheres to the highest substantive and formal standards.

**Description Of Results:**

5: Very High Quality – The results are described in detail, clearly and comprehensively, supported by thorough evaluation, analysis, and convincing usage examples. The description meets the highest substantive standards.

**Feedback On Consistency:**

Praca jest spójna, logiczna i dobrze opisana.

**Potential For Development:**

W artykule wskazano dalsze możliwe kierunki rozwoju projektu.

**Project Nature Evaluation:**

Projekt wykazuje cechy pracy inżynierskiej.

**Technical Language Precision:**

5: Very High Quality – The language is entirely appropriate for a technical report. All terms are used correctly and precisely, and the style is professional, clear, and coherent, without any errors or ambiguities.

---

### Official Review · Reviewer_Lfmr · 2024-12-06
**A review of an online system for image editing using AI models**

**Confidence:** 5
**Significance Of Results:** 4
**Overall Quality:** 4

**Compliance With Template:**

5: Very High Quality – The article contains all the required sections, which are written in a very detailed, clear, and error-free manner. The structure is professional and meets expectations, and the content adheres to the highest substantive and formal standards.

**Description Of Results:**

5: Very High Quality – The results are described in detail, clearly and comprehensively, supported by thorough evaluation, analysis, and convincing usage examples. The description meets the highest substantive standards.

**Feedback On Consistency:**

The problem description consistency is high enough. Authors present the project technical details and the describe the AI models applied. The system architecture is described as well as the project objectives.
The commercialization potential is high.

**Potential For Development:**

The authors point out the possible development of the application by adding new AI solutions.

**Project Nature Evaluation:**

This is an engineering work. The project technical details are  are presented as well.

**Technical Language Precision:**

4: High Quality – The language is appropriate for a technical report. Terminology is used correctly, and statements are precise, with only minor shortcomings that do not affect the overall clarity.

---

### Decision · Program_Chairs · 2024-12-10

Accept (Poster)